CellPress

## Perspective

# Equity, diversity, and inclusion at the Global Alliance for Genomics and Health

Neerjah Skantharajah,[1,2] Shakuntala Baichoo,[3] Tiffany F. Boughtwood,[4,5] Esmeralda Casas-Silva,[6]
Subhashini Chandrasekharan,[7] Sanjay M. Dave,[8] Khalid A. Fakhro,[9,10] Aida B. Falcon de Vargas,[11,12] Sylvia S. Gayle,[6]
Vivek K. Gupta,[13] Rachele Hendricks-Sturrup,[14] Ashley E. Hobb,[15] Stephanie Li,[2,16] Bastien Llamas,[17,18,19,20]
Catalina Lopez-Correa,[21] Mavis Machirori,[22,23] Jorge Melendez-Zajgla,[24] Mareike A. Millner,[25] Angela J.H. Page,[2,16]
Laura D. Paglione,[26,27] Maili C. Raven-Adams,[2,28] Lindsay Smith,[1,2] Ericka M. Thomas,[7] Judit Kumuthini,[29]
and Manuel Corpas[30,*]

[1]Ontario Institute for Cancer Research, Toronto, ON, Canada
[2]Global Alliance for Genomics and Health, Toronto, ON, Canada
[3]University of Mauritius, Reduit, Mauritius
[4]Australian Genomics, Parkville, VIC, Australia
[5]Murdoch Children's Research Institute, Parkville, VIC, Australia
[6]National Cancer Institute, Rockville, MD, USA
[7]The All of Us Research Program, National Institutes of Health, Bethesda, MD, USA
[8]Department of Biotechnology, Hemchandracharya North Gujarat University, Patan, Gujarat, India
[9]Department of Human Genetics, Sidra Medicine, Doha, Qatar
[10]Department of Genetic Medicine, Weill Cornell Medical College, Doha, Qatar
[11]Hospital Vargas de Caracas, Vargas Medical School, Universidad Central de Venezuela, Caracas, Venezuela
[12]Hospital de Clínicas Caracas, Caracas, Venezuela
[13]Macquarie Medical School, Faculty of Medicine, Health and Human Sciences, Macquarie University, Sydney, NSW, Australia
[14]Duke-Margolis Center for Health Policy, Washington, DC, USA
[15]DNAstack, Toronto, ON, Canada
[16]Broad Institute, Cambridge, MA, USA
[17]Australian Centre for Ancient DNA, School of Biological Sciences and The Environment Institute, University of Adelaide, Adelaide, SA, Australia
[18]ARC Centre of Excellence for Australian Biodiversity and Heritage, University of Adelaide, Adelaide, SA, Australia
[19]National Centre for Indigenous Genomics, John Curtin School of Medical Research, Australian National University, Canberra, ACT, Australia
[20]Indigenous Genomics, Telethon Kids Institute, Adelaide, SA, Australia
[21]Genome Canada, Ottawa, ON, Canada
[22]Ada Lovelace Institute, London, UK
[23]PEALS, Newcastle University, Newcastle Upon Tyne, UK
[24]Instituto Nacional de Medicina Genomica, Mexico City, Mexico
[25]Maastricht University, Health Law and Governance Group, Maastricht, the Netherlands
[26]Spherical Cow Group, New York, NY, USA
[27]Laura Paglione LLC, New York, NY, USA
[28]Wellcome Sanger Institute, Hinxton, UK
[29]South African National Bioinformatics Institute, University of Western Cape, Cape Town, South Africa
[30]School of Life Sciences, University of Westminster, London, UK
*Correspondence: m.corpas@westminster.ac.uk

## SUMMARY

A lack of diversity in genomics for health continues to hinder equitable leadership and access to precision medicine approaches for underrepresented populations. To avoid perpetuating biases within the genomics workforce and genomic data collection practices, equity, diversity, and inclusion (EDI) must be addressed. This paper documents the journey taken by the Global Alliance for Genomics and Health (a genomics-based standard-setting and policy-framing organization) to create a more equitable, diverse, and inclusive environment for its standards and members. Initial steps include the creation of two groups: the Equity, Diversity, and Inclusion Advisory Group and the Regulatory and Ethics Diversity Group. Following a framework that we call "Reflected in our Teams, Reflected in our Standards," both groups address EDI at different stages in their policy development process.

## INTRODUCTION

While individuals of European descent represent approximately 16% of the global population, the overwhelming majority of research data sources originate from people of European ancestry.[1–3] Research studies that do include underrepresented groups risk being disregarded by scientists due to their low statistical power,[4] resulting in genomic research and associated medical practices being informed by and benefitting only a fraction of the global population.[5]

The lack of diversity in human genomics research extends beyond research participants. For example, while women constitute over 50% of the biomedical science undergraduate and postgraduate population, they only account for 18% of biomedical science professors.[6] Lack of diversity in the workforce can further perpetuate the lack of underrepresented groups and communities in research studies. As a result, this may bias scientific research questions and methodologies, affecting research results and their potential applicability across diverse populations.[7–9] Indeed, there have been increasing calls for diversity in the genomics workforce by research funders.[10]

The results of homogeneous representation of research participants and the research workforce are limiting.[11] Genomic-based medical practices may be limited in addressing health issues in specific groups and communities, yielding limited or skewed data results in biased algorithms and devices.[12] Equity, diversity, and inclusion (EDI) practices are thus needed in the genomics field both in terms of patient data, research participation, and workforce representation to better understand the nature and manifestation of global variation in genetic data and implications for health.

## STANDARDS FOR EQUITY

In the development phase of standards, global participation is key to ensuring the standards produced address the needs of researchers, research participants, and patients. Equity in this context thus refers to the fair and just distribution of benefits and opportunities related to genomic healthcare.[13] Standards for equity in genomics for health may therefore provide a framework for comparing and evaluating access to genomic information. This in turn would enable fair opportunities for communication and collaboration among researchers, healthcare providers, and patients.[14] Standards may also help establish interoperability and consistency across different aspects of genomics, including data generation, sequencing technologies, bioinformatics pipelines, variant interpretation, and reporting.

### Standards for diversity

In our context, diversity refers to representation of difference in aspects including, but not limited to, race, ethnicity, ancestry, geography, religion, sex, sexual orientation, gender identity/expression, disability, age, traditions, language, dialect, and socio-economic and cultural factors. Standards facilitating the collection of genomic data from diverse populations may help promote the accuracy and applicability of genomic information across broader populations.[15] These standards could in turn enable more accurate diagnosis, treatment, and risk assessment for individuals from different backgrounds.

## UNDERSTANDING THE NATURE OF THE PROBLEM

The Global Alliance for Genomics and Health (GA4GH), a genomics standards-setting body, was built in January 2013 on the foundational human right to benefit from science and its applications[16]; however, this right cannot be truly realized until issues related to EDI are identified and addressed. We present GA4GH's journey to date in understanding and responding to EDI-related responsibilities and challenges. Our approach is one that we hope will promote dialogue and encourage concerted action among similar organizations in genomics, health, and standards development communities.

GA4GH is dedicated to the acceleration of progress in genomics research and human health. It achieves this by harmonizing approaches for effective and responsible sharing of genomic data through standard setting and policy framing. These standards and policies are developed by GA4GH's "Work Stream" groups. Each Work Stream represents a problem space within the genomics field (e.g., data security, large-scale genomics) and may also have subgroups that work toward the development of specific products. However, despite boasting an impressive 600+ organizational members, 1,000+ Work Stream members, and 300+ contributors that participate across these groups, EDI metrics do not reflect the diversity and richness of the constituency GA4GH is seeking to represent.

### Organizational EDI measurements

An August 2021 internal audit revealed that geographically, 32 countries are represented among GA4GH's volunteers, but only 22% of members reside outside of the Anglosphere (Australia, Canada, New Zealand, the United Kingdom, and the United States).[17] Among Work Stream and subgroup leads, five of the nine countries represented are not within the Anglosphere, but members residing in these countries only make up 7% of the group. Moreover, 18% of leads identify as female. When looking at GA4GH's Steering Committee, which is responsible for making high-level decisions regarding direction, values, and deliverables, there are 11 countries represented with 38% of members residing in non-anglophone countries. Among the Steering Committee members, 21% of the committee identify as female. Figure 1 provides a visual breakdown of GA4GH's composition. Further diversity metrics are limited as they either have not been measured (e.g., race, ethnic origin, religion) or have only been measured for a subset of GA4GH members (e.g., gender, place of origin).

## OUR JOURNEY INTO ADDRESSING EDI BIAS

GA4GH's first step toward addressing prevailing EDI issues was the creation of two separate and complementary working groups. The EDI Advisory Group was developed focusing on bringing diverse ideas into the standards creation process with a specific focus on onboarding and participation levels of members. The Regulatory and Ethics Work Stream (REWS) Diversity Group was initiated to ensure that the standards themselves and associated development processes are carried out in an equitable manner.[18] Together, these two groups aim to address EDI issues within GA4GH following the "Reflected in our Teams,

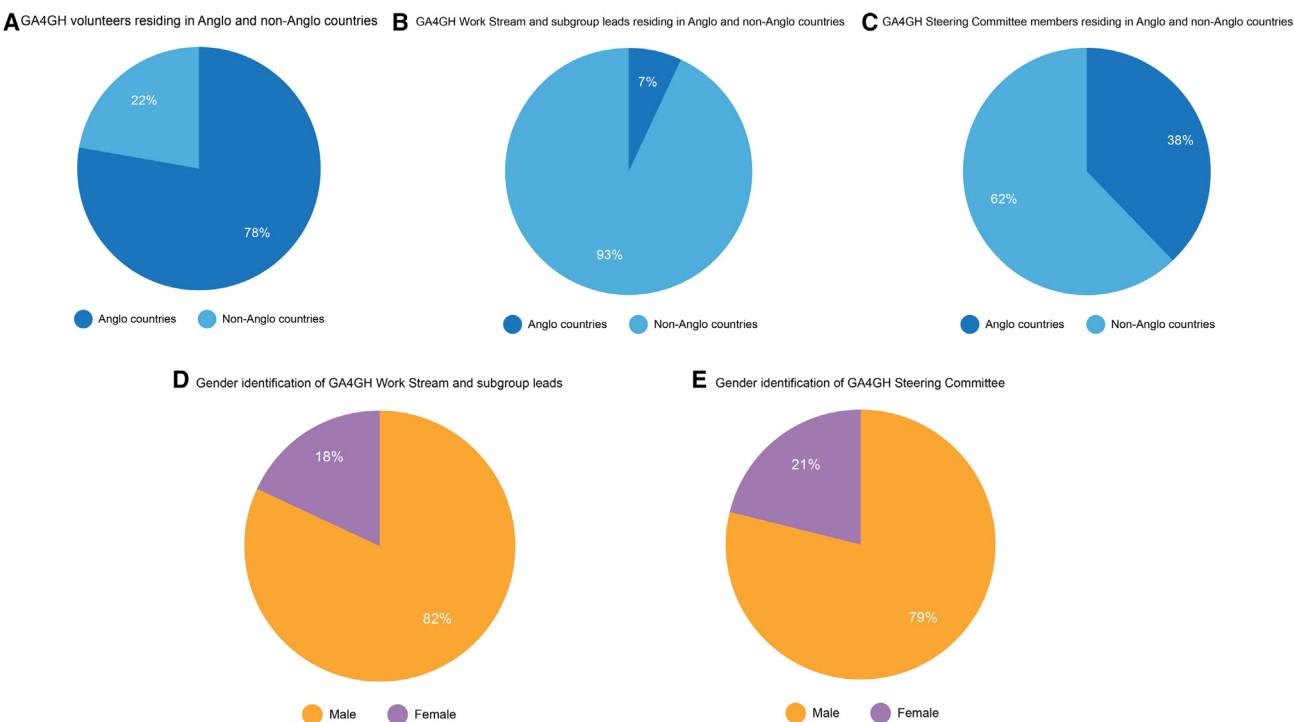

**Figure 1. Visual breakdown of the representation found within GA4GH volunteers, leads, and steering committee members**

Reflected in our Standards'' framework. This framework is based on the idea that if EDI is incorporated successfully into our teams, then those considerations will be reflected in the standards development process and in the standards themselves. This framework and its implications are further described in Figure 2.

At implementation level, global participation remains crucial as products are altered to reflect the implementers' language, context, and culture. For example, the GA4GH Data Use Ontology, a standard that allows users to tag genomic datasets with usage restrictions semantically, was translated into Japanese, French, Spanish, and German to facilitate international uptake.[19] However, a suitable technical framework supporting the ongoing, consistent release of translations, and their update alongside the original standard when it evolves, remains to be implemented. Additionally, we expect the inclusion of languages spoken in the global south will increase adoption of the standard further.

## GA4GHs EDI WORKING GROUPS

The GA4GH's first expedition into EDI work began in December 2014, prior to the creation of the EDI Advisory and REWS Diversity groups. At this time, the REWS published the *Framework for Responsible Sharing of Genomic and Health-Related Data*.[20] The framework is guided by Article 27 of the 1948 Universal Declaration of Human Rights.[21] Article 27, the right to cultural, artistic, and scientific life, posits that all individuals have the right to ''to share in scientific advancement and its benefits.'' Initially published in English, the framework has been foundational to

GA4GH and its activities and is now translated in 14 additional languages.

The REWS published GA4GH's *Framework for Involving and Engaging Participants*, *Patients and Publics in Genomics Research and Health Implementation*,[22] which speaks to a commitment to promote and use diversity to improve genomics initiatives. It further suggests that gaining trust and establishing transparency and accountability in science matter, as these can negatively or positively affect everyone engaged. Therefore, these considerations are necessary for scientific and research practice and in order to uphold the principles of beneficence and non-maleficence as a moral and ethical obligation in scientific conduct within the scope of human health.

## EDI Advisory Group

The EDI Advisory Group was launched in May 2020 with the goal of recognizing and responding to EDI issues raised within the GA4GH community, such as a lack of diversity in Work Streams, leadership, and conferences. This group supports GA4GH by finding equitable and inclusive ways to bring diverse ideas into the standards creation process. Pathways to EDI that the group explores include the following:

(1) identifying opportunities to attract diverse talents to the GA4GH community;
(2) helping new and existing contributors from all backgrounds feel welcomed and valued by the community;
(3) ensuring equitable access to leadership and speaking opportunities within GA4GH.

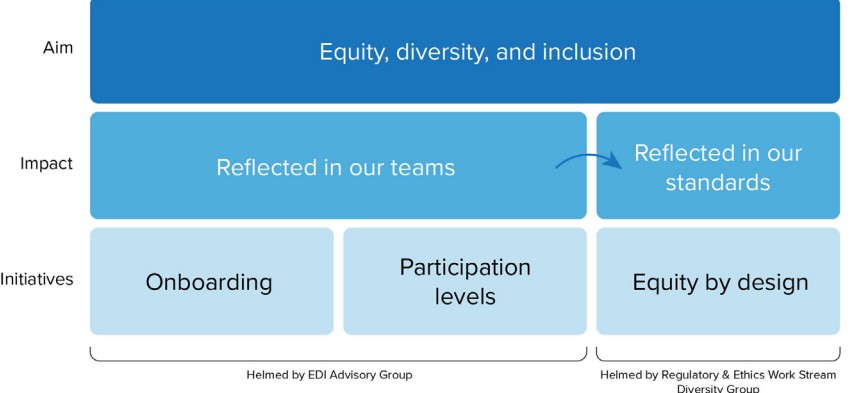

Figure 2. Mapping the "Reflected in our Teams, Reflected in our Standards" framework to specific target areas that will be tackled by the EDI Advisory Group and REWS Diversity Group
All initiatives aim to achieve equity, diversity, and inclusion.

Work of this group is closely coordinated with the Secretariat and across GA4GH Work Stream activities to ensure approaches and deliverables can ultimately feed back into the community.

The EDI group has examined the community pillars as defined by Vogl,[23] which can reinforce the feeling of community, amplify community participation, and increase the sense of belonging. These pillars are shown in Table 1.

### REWS Diversity Group
The genesis of the REWS Diversity subgroup was informed by the notion that EDI concerns should be addressed early and often in the development process and not at the end.

The first output from REWS Diversity has been the living definition of EDI for REWS diversity that was developed in collaboration with the EDI Advisory Group. The definition is based on the understanding that in EDI are three interrelated concepts that begin with the absence of exclusion and are founded upon an ethos of intention and advocacy. "Equity" is defined as the removal of systemic barriers and biases enabling all individuals to have just and fair opportunities to access and benefit from scientific advancement. Benefitting from scientific advancement entails both promoting science while mitigating potential harms that scientific research creates. In this context, equity is a value embedded within dignity and the human right to benefit from science and, thus, a value implicitly recognized in GA4GH's *Framework for responsible sharing of genomic and health-related data*.[24] "Diversity" is defined as the embrace of variations and differences in genetic characteristics, race, color, place of origin, geography, religion, immigrant and newcomer status, ethnic origin, ability, sex, sexual orientation, gender identity, gender expression, age, and other socio-economic and cultural factors. "Inclusion" ensures that all individuals are welcomed and accepted, equitably supported and empowered, and that their contributions are valued and respected, and it requires proactive and intentional practice.

### LIMITATIONS AND CHALLENGES

Various factors limit the ability of diverse groups to engage in organizations such as GA4GH. These include but are not limited to a lack of time or resources for meaningful engagement; an inability to network with the members of the community due to social, political, structural, and/or language related barriers; practical constraints of meeting scheduling across time zones; a lack of awareness of this work; and an impression that this work is not designed for one's population.

Individuals who do not have access to a stable internet connection, or the technology to access it, are also at a disadvantage. This subsequently limits individuals' ability to meaningfully engage and participate in collaborative work and can create underrepresentation and imbalance in more visible types of engagement such as leadership and speaking opportunities. These amplified inequities can be potentially addressed by diversifying communication channels when possible. These limitations can also potentially be addressed by instituting awards and reserving board positions for underrepresented groups.

Within the genomics community, there is a recognition of the need for diversity in participants, data, and the workforce.[25] What remains unclear are the standards/methods that can be used to support engagement with diverse communities, which can be mitigated by organizations in the genomics and health landscape sharing their approaches for engaging individuals from diverse backgrounds. It is also important to enlist grass-root organizations with practical strategies for inclusion but may not publish through peer-reviewed literature. Countries outside the Anglosphere must be given a voice in these discussions to inform culturally appropriate and successful engagement within their communities. Creating a culture of collaboration where groups working toward the same goal can share both their successes and failures will accelerate the communities' advancement toward diversity and inclusion.

Evaluation of EDI initiatives is therefore essential to understanding their impact on the community, but recording measurements within an open community remains a challenge. If not addressed, organizations like GA4GH run the risk of operating in silos and/or perpetuating known and unknown biases within the genomics workforce and through leadership as well as within genomic data collection practices, policies, and infrastructures.

Despite the benefits of diversity in producing high-quality work products, collaborative groups are composed of individuals with uneven capacities to volunteer for GA4GH. Creating a rich and engaging participation culture will be important not only for engagement but also to maintain long-lasting relationships with diverse communities.

**Table 1. The community pillars adopted by the EDI Advisory Group**

| No. | Community pillar | Definition |
|---|---|---|
| 1 | boundaries | a clear and shared understanding of who is included in the core of this community |
| 2 | rituals | identification and formalization of the things we do as a community and how we involve members of the community in these activities |
| 3 | home | specific physical and virtual locations where one can go to engage with the community |
| 4 | stories | values and stories that members of the community know and share |
| 5 | symbols | descriptions of specific things that represent the ideas that are important to the community |
| 6 | participation levels | clear path for increased engagement in the community as one participates |

The need to focus our early EDI efforts on areas where there was both the greatest need and the highest potential for positive impact on our intentional community was agreed. Through community discussions, surveys, and workshops, the group has identified potential high impact projects to address EDI within GA4GH (Table 2).

## THREE CALLS TO ACTION

Developing comprehensive policies that address disparities, promote diversity, and ensure equitable access to genomic data can be a complex process, hindered by bureaucratic challenges and differing priorities.

Addressing these barriers necessitates a multi-faceted approach involving collaboration between researchers, policymakers, healthcare providers, and community advocates. Efforts are required to increase awareness, promote diversity in research, secure funding, improve technological infrastructure, address ethical concerns, and develop supportive policies to overcome these barriers and to facilitate the implementation of equity standards. Table 3 shows our three calls to action toward creating equitable, diverse, and inclusive environments that intentionally include diverse individuals. These are grounded on (1) sharing learnings, (2) assessing EDI culture, and (3) promoting work at the intersection of EDI and genomics.

Consensus-driven codes, policies, and frameworks within the GA4GH community importantly serve to guide researchers and practitioners through personal and cultural self-reflection and action toward better practices. GA4GH's acknowledgment, commitment, and actions to set individual and research community examples and standards are critical, but real change will seldom occur if the change model is not broadly adopted in practice, which would affirmatively drive that change forward.

**Table 2. Names and activities of projects that aim to encourage equity, diversity, and inclusion within GA4GH**

| Project name | Activity |
|---|---|
| Work Stream Meeting Best Practices | • introduce Work Stream Leads to best practices for creating an inclusive environment: promoting introductions at the start of meetings<br>• reassessing GitHub culture and the level of technical language used<br>• considering cultural differences in discussions<br>• providing context at the start of each meeting<br>• adding contextual information to documents such as its purpose, and how individuals should interact with it |
| EDI Analysis of GA4GH Role Structure/Barriers | • evaluate the impact of having multiple levels of hierarchy<br>• identify effective methods of communicating the nature of GA4GH processes |
| Genomics-centered EDI Definition | • creating a definition of EDI that is specific to genomic standards to serve an educational purpose and inform future work |
| Website Redesign | • identify/reduce barriers caused by GA4GH-related jargon<br>• providing more information on how decisions are made at GA4GH |
| Leadership Selection Pool | • targeted outreach to identify experts from different countries and backgrounds<br>• opening calls for candidacy to the broader community rather than only internal candidates |
| Recognizing the Groups Participating in Standards | • explicitly recognize the geographic areas/income levels in internal and external documents that were considered and/or the expertise of the working group who created a standard |
| Mentorship within Driver Project and Work Streams | mentorship provision by longtime contributors to the GA4GH community to support:<br>• newcomers to the community as they are onboarded<br>• individuals who could become candidates for leadership roles, e.g., Work Stream Lead |

Projects have been developed by the EDI Advisory Group.

**Table 3. Three calls to action to help guide members of genomics research communities**

| No. | Call to action |
|---|---|
| 1 | "share" learnings about how to effectively identify and address EDI needs |
| 3 | "assess" and discuss EDI culture and practices |
| 3 | "promote" work at the intersection of EDI and genomics |

These are designed to promote equitable, diverse, and inclusive environments that intentionally include diverse individuals.

## Conclusion

Better EDI practices can increase innovation, creativity, and impact.[26–28] We envision a future where an inclusive GA4GH community creates effective standards that work across a spectrum of contexts, ultimately impacting the public at large. We foresee a collaborative GA4GH that welcomes people of all backgrounds, with a range of professional expertise and from various disciplinary fields to engage and fully participate.

The observations explored in this paper are GA4GH's steps toward responding to EDI concerns within the community. We expect the work of these groups to broaden participation of members from diverse groups and, consequently, to lead to the creation of more effective and far-reaching data sharing and leadership standards.

A key aspect of this journey will be understanding what success looks like in order to actively and regularly evaluate our progress on EDI. As the groups continue to implement changes, develop initiatives, and produce policies, it will be essential to incorporate evaluation of GA4GH's level of success. Examples of metrics that GA4GH commits to tracking include diversity metrics as a function of participation levels and landscape mapping of how problems are approached and addressed through an EDI lens. More specifically, the EDI groups have begun participating in active discussion around how to measure diversity metrics more regularly and accurately for a larger proportion of active GA4GH members.

Our proposed three calls to action will be instrumental in helping to drive positive change across the GA4GH organization. These include (1) sharing learnings, (2) assessing EDI culture, and (3) promoting work at the intersection of EDI and genomics for health. It is our hope that as we, and other related organizations, respond to our three calls, we begin to see a scientific community that is more representative and fairer for the global community that we aim to serve.

## SUPPLEMENTAL INFORMATION

## ACKNOWLEDGMENTS

B.L. acknowledges funding by Australian Research Council Future Fellowship (FT170100448); A.J.H.P. acknowledges funding from the Broad Institute; B.L. acknowledges funding from the Future Fellowship of the Australian Research Council (FT170100448); T.B. acknowledges funding from the National Health and Medical Research Council (NHMRC), Australia (grant GNT2000001); V.K.G. acknowledges funding from the National Health and Medical Research Council (NHMRC), Australia; J.K. acknowledges funding from National Human Genome Research Institute (NHGRI) grant number U24HG006941; E.M.T. and J.K. acknowledge funding from the National Institutes of Health; L.S. and N.S. acknowledge funding from the National Institutes of Health (grant U24HG011025); L.S. and N.S. acknowledge funding from Ontario Institute for Cancer Research and the Canadian Institutes of Health Research. We acknowledge all current and past members of the GA4GH REWS Diversity and EDI Advisory Groups for their contributions toward frameworks, initiatives, policies, and other products discussed in this paper. The views expressed in this manuscript are those of the authors and do not necessarily represent the views of the institutions with which each individual is affiliated.

## AUTHOR CONTRIBUTIONS

Conceptualization, A.E.H., C.L., E.C., J.K., J.M., K.A.F., L.D.P., M.C., N.S., T.F.B., and V.K.G.; funding acquisition, A.E.H.; investigation, L.D.P. and M.A.M.; methodology, A.E.H. and K.A.F.; project administration, A.E.H., J.K., L.D.P., and N.S.; resources, L.D.P.; supervision, J.K., L.D.P., and M.C.; visualization, S.L.; writing – original draft, J.K., K.A.F., M.C., N.S., S.B., S.L., S.S.G., and V.K.G.; writing – review & editing, A.B.F., A.E.H., A.J.H.P., B.L., E.C., E.M.T., J.K., J.M., K.A.F., L.D.P., L.S., M.A.M., M.C.R., M.C., M.M., N.S., R.H.S., S.B., S.C., S.M.D., T.F.B., and V.K.G.

## DECLARATION OF INTERESTS

As of this writing, M.C. is associated with Cambridge Precision Medicine Limited. R.H.-S. is associated with the National Alliance Against Disparities in Patient Health.

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
