## [Document S1. Transparent peer review records for Skantharajah et al · Cell Genomics]

Cell Genomics, Volume 3

Supplemental information

Equity, diversity, and inclusion

at the Global Alliance for Genomics and Health

Neerjah Skantharajah, Shakuntala Baichoo, Tiffany F. Boughtwood, Esmeralda Casas-Silva, Subhashini Chandrasekharan, Sanjay M. Dave, Khalid A. Fakhro, Aida B. Falcon de Vargas, Sylvia S. Gayle, Vivek K. Gupta, Rachele Hendricks-Sturup, Ashley E. Hobb, Stephanie Li, Bastien Llamas, Catalina Lopez-Correa, Mavis Machirori, Jorge Melendez-Zajgla, Mareike A. Millner, Angela J.H. Page, Laura D. Paglione, Maili C. Raven-Adams, Lindsay Smith, Ericka M. Thomas, Judit Kumuthini, and Manuel Corpas

Equity, Diversity and Inclusion at the Global Alliance for Genomics and Health

Author list

Neerjah Skantharajah, Shakuntala Baichoo, Tiffany F. Boughtwood, Esmeralda Casas-Silva, Subhashini Chandrasekharan, Sanjay M. Dave, Khalid A. Fakhro, Aida B. Falcon de Vargas, Sylvia S. Gayle, Vivek K. Gupta, Rachele Hendricks-Sturup, Ashley E. Hobb, Stephanie Li, Bastien Llamas, Catalina Lopez-Correa, Mavis Machirori, Jorge Melendez-Zajgla, Mareike A. Millner, Angela J.H. Page, Laura D. Paglione, Maili C. Raven-Adams, Lindsay Smith, Ericka M. Thomas, Judit Kumuthini, Manuel Corpas

Summary

Initial submission: Received : March 4th 2023

Scientific editor: Judith Nicholson

First round of review: Number of reviewers: 2
Revision invited : May 3rd 2023
Revision received : Jun 24th 2023

Second round of review: Number of reviewers: N/A
Accepted : 1st August 2023

Data freely available: N/A

Code freely available: N/A

This transparent peer review record is not systematically proofread, type-set, or edited. Special characters, formatting, and equations may fail to render properly. Standard procedural text within the editor's letters has been deleted for the sake of brevity, but all official correspondence specific to the manuscript has been preserved.

Referees' reports, first round of review

Reviewer 1

For starters - I really appreciate the authors wanting to document a journey of thinking/framing/positioning and logic in setting up these groups. Often the underlying assumptions are not well articulated and therefore it can be hard to understand why, maybe such groups end up missing key issues or not being as effective as possible. However, at the moment the paper falls short of being as useful and interesting as it can be. I feel that if it positioned itself as "Here's a way to approach setting up group to address and interview on EDI matters - These are the questions you might want to consider - here's what we did (not saying this is a gold standard), we've broken it down into simple steps/references to key resources we used as a worked example/guide, here's what we are going to be keeping an eye on", then it could become a lot more useful for the wider community. This could also be more aided by lists/steps in a process (with full transparency that you probably didn't set out the blocks with a rigid process in mind, but looking back you set the path for x,y,z steps) as at the moment the structure is lacking in clarity.

Potential structure (lots of it is there already)

1) Data gathering/understanding the nature of the problem in GA4GH

2) Building thoughtful foundations

- Key considerations:

- Section on the relationship between standards and equity (with referenced/evidenced examples)

- Section on how more diversity in collaborative/deliberative/coproduced standards is better e.g. Evidencing and further enriching statements like "A broader perspective from the outset would lead to earlier recognition of potential challenges

- Measurements of community health/effectiveness e.g.

<https://chaoss.community/kb-metrics-and-metrics-models/> and/or different ways

to consider the "health" of a community or working group
- Others?

3) Deep dive into the EDI working groups

Great section showing the journey - how did you get from those pillars to those projects? Going from problems to solutions is a big jump so it would be good to understand the origin of that great list of projects. Benefit from a process diagram here I suspect.

4) Limitations

At the moment this is focussed on engagement, I think you could go further and maybe map the key types of limitations/barriers for GA4GH's effectiveness (as they related to equity).

Specific minor comments

- Second sentence of the abstract is grammatically clunky, but overall the abstract needs to be reworked if it takes more of a "guide + example" format
- 78% GWAS stat - whilst I agree this is very good example, it's such an overused example, if a list of 3 representative examples or a different headline could be given it would be good
- Language like devastating and bleak aren't awesome
- "In this paper, diversity refers to characteristics including..." - Seems unusual that ancestry is not in the list given that's the main one

Overall, a promising paper - as a standards-setter, the authors are potentially missing a trick in terms of how practical the paper is/how compelling the journey is

Reviewer 2

The authors present an overview of the processes being put in place by GA4GH to address equity, diversity and inclusion. Clearly a timely and important issue in genomic medicine, and overall the paper provides a very useful summary of a large amount of work that GA4GH has put into considering this problem. Overall I think the paper is great, but I do find it overly wordy and repetitive in some areas. I think it could be tightened up and bolstered with some more specific examples to make this something the journal readership would find more helpful and

informative.

- The introduction is quite long and some aspects of this are then again broadly repeated in the main body of the paper. Suggest some attention be made to trimming and tightening up the paper (especially the introduction), which will also make the key points here punchier and have greater impact.
- Equity, diversity and inclusion are not defined until much further on in the paper (after the very long introduction). I think this needs to come earlier to increase clarity. It does feel like those terms are used interchangeably throughout the introduction, which of course becomes clear that is not the intent much later.
- Background, page 2: insufficient statistical power is quoted as a barrier for publication by under-represented groups. I suspect there is also publication bias in that well known groups find it easier to publish in higher impact journals, among other barriers. It feels like this description is overly simplistic written as it is?
- Many diverse groups are considered here, should career stage also be included?
- How do people become involved in GA4GH?
- Is anglosphere a commonly used term? Is the point that these are high income countries of largely white/European background?
- The paper is reporting on outcomes of workshops and consultation etc, but I think there is a risk of speaking in such broad terms that it seems you don't consider the challenges in achieving EDI goals. I would find the paper considerably more useful if some increased specificity could be added, such as specific examples of what some of these proposed steps might be. I think this would then also allow for some consideration of the challenges of EDI goals to be discussed. I think this would take this from being a very general, "we need to be more equitable" paper to something that attempts to provide real guidance.
- Likewise, some of the suggested steps, like "Engage in outreach practices" (page 9) are easy to suggest but much more difficult to implement. As above, more specificity about how the groups think this might actually be achievable could be really powerful.
- Given the difficulties, there is very brief mention of how EDI goals might be evaluated, but considering the significant investment in time/resources being proposed (because truly achieving equity, diversity and inclusion really does require significant effort), surely these are a key part of such working group activities?

Authors' response to the first round of review

Below we explain in red the answers to both reviewers individually, indicating the changes we have performed in the manuscript, thanking them again for their valuable comments.

REVIEWER 1:

For starters - I really appreciate the authors wanting to document a journey of thinking/framing/positioning and logic in setting up these groups. Often the underlying assumptions are not well articulated and therefore it can be hard to understand why, maybe such groups end up missing key issues or not being as effective as possible. However, at the moment the paper falls short of being as useful and interesting as it can be. I feel that if it positioned itself as "Here's a way to approach setting up group to address and interview on EDI matters - These are the questions you might want to consider - here's what we did (not saying this is a gold standard), we've broken it down into simple steps/references to key resources we used as a worked example/guide, here's what we are going to be keeping an eye on", then it could become a lot more useful for the wider community. This could also be more aided by lists/steps in a process (with full transparency that you probably didn't set out the blocks with

a rigid process in mind, but looking back you set the path for x,y,z steps) as at the moment the structure is lacking in clarity.

ANSWER:

We would like to thank Reviewer 1 for recognizing the potential of this paper and for the thinking that they have put into making it more readable for a wider audience. We have taken great care to carefully consider their suggestions and have reworked the manuscript extensively. There has been a significant rearrangement of content (including rewording of headings) as well as addition of sections. We hope that the changes in the structure and the content improves the flow of the text, with the aim of particularly emphasizing the nature of the problem, the journey and measures taken, and specific lists / steps suggested in order to make it of greater interest to a wider audience. We elaborate below on the changes to the structure as suggested by Reviewer 1.

Potential structure (lots of it is there already)

1) Data gathering/understanding the nature of the problem in GA4GH

2) Building thoughtful foundations

- Key considerations:

- Section on the relationship between standards and equity (with referenced/evidenced examples)

- Section on how more diversity in collaborative/deliberative/coproduced standards is better e.g. Evidencing and further enriching statements like "A broader perspective from the outset would lead to earlier recognition of potential challenges

- Measurements of community health/effectiveness e.g.

<https://chaoss.community/kb-metrics-and-metrics-models/> and/or different ways to consider the "health" of a community or working group

- Others?

3) Deep dive into the EDI working groups

Great section showing the journey - how did you get from those pillars to those projects? Going from problems to solutions is a big jump so it would be good to understand the origin of that great list of projects. Benefit from a process diagram here I suspect.

4) Limitations

At the moment this is focussed on engagement, I think you could go further and maybe map the key types of limitations/barriers for GA4GH's effectiveness (as they related to equity).

ANSWER:

We changed the structure of the paper to reflect the one suggested by Reviewer #1. We now have the major headings of 'Introduction', 'Understanding Nature of the Problem', 'Our Journey into Addressing EDI Bias', 'GA4GH's EDI Working Groups', 'Limitations and Barriers' and 'Three Calls to Action'. We have also remodeled the inside of the above headings in order to provide more bite-sized content. For instance, in Introduction, we have added two subsections that set the scene for the definition of our work and policies: 'Standards for Equality' and 'Standards for Diversity'. These two sections within the introduction evidence the relationship between standards and equity and how more diversity in collaborative/deliberative/coproduced standards is better (as suggested in (2) for building thoughtful foundations).

Within the 'Nature of the Problem' we have now have added a subsection on 'Organization EDI Measurements' that refers to the measurement of the community's health/effectiveness.

The section on 'Our Journey into Addressing EDI Bias' suggests the origins of the EDI Advisory Group and the Regulatory Work Steam Diversity Group. These two groups constitute two parallel streams from which projects and actions have sprouted to address EDI biases.

We have created a new table, Table 1, where we list the pillars from which such projects and activities emanate for our EDI Advisory Group. We believe that this table will be of wide interest since they are not specific to GA4GH, but communities in general, and the fact that we make these pillars into a table also adds to the suggested need to add more steps and lists into the paper.

Next, we go into deep detail describing the foundations for the establishment of the EDI Advisory Group, for which we then show its different projects and derived activities in Table 2.

After we have described the living Definition of EDI for REWS Diversity, which is their main output, we go into 'Limitations and Barriers'. Here we specify the key types of limitations for this work, including a lack of time or resources for meaningful engagement; an inability to network with the members of the

community due to social, political, structural, and/or language related barriers; practical constraints of meeting scheduling across time zones; a lack of awareness of this work; and an impression that this work is not designed for one's population.

Finally, we have created Table 3 to emphasize clarity on steps towards adoption of EDI awareness and practice through our three calls to action. We have changed the section's title into 'Three Calls to Action' as a 'take away / take home message' for our paper.

Specific minor comments

- *Second sentence of the abstract is grammatically clunky, but overall the abstract needs to be reworked if it takes more of a "guide + example" format*
- *78% GWAS stat - whilst I agree this is very good example, it's such an overused example, if a list of 3 representative examples or a different headline could be given it would be good*
- *Language like devastating and bleak aren't awesome*
- *"In this paper, diversity refers to characteristics including..." - Seems unusual that ancestry is not in the list given that's the main one*

Overall, a promising paper - as a standards-setter, the authors are potentially missing a trick in terms of how practical the paper is/how compelling the journey is

ANSWER:

- We have reworked the abstract to make it more streamlined.
- We have eliminated the 78% stat and focused on the small proportion of population in the world that is of European origin, adding 3 references to the statement.
- We have deleted/edited out 'bleak' and 'devastating'
- We have included 'ancestry' as a component of diversity

REVIEWER 2:

The authors present an overview of the processes being put in place by GA4GH to address equity, diversity and inclusion. Clearly a timely and important issue in genomic medicine, and overall the paper provides a very useful summary of a large amount of work that GA4GH has put into considering this problem.

Overall I think the paper is great, but I do find it overly wordy and repetitive in some areas. I think it could be tightened up and bolstered with some more

specific examples to make this something the journal readership would find more helpful and informative.

ANSWER:

Thank you for your comment and appreciation. We have undertaken a major editing process to make it less repetitive and tighten the language. We have also aimed at adding more specific examples and references which we explain below in detail. A lot of what we have done to satisfy this reviewer's comments has been to actually edit out some of the content that on reflection was too unspecific.

- The introduction is quite long and some aspects of this are then again broadly repeated in the main body of the paper. Suggest some attention be made to trimming and tightening up the paper (especially the introduction), which will also make the key points here punchier and have greater impact.

ANSWER:

We have significantly trimmed our Introduction in this revised version to half a page.

- Equity, diversity and inclusion are not defined until much further on in the paper (after the very long introduction). I think this needs to come earlier to increase clarity. It does feel like those terms are used interchangeably throughout the introduction, which of course becomes clear that is not the intent much later.

ANSWER:

In Introduction, we have added two subsections that set the scene for the definition of our work and policies: 'Standards for Equality' and 'Standards for Diversity'.

- Background, page 2: insufficient statistical power is quoted as a barrier for publication by under-represented groups. I suspect there is also publication bias in that well known groups find it easier to publish in higher impact journals, among other barriers. It feels like this description is overly simplistic written as it is?

ANSWER:

We have edited out the point of insufficient statistical power in page 2.

- Many diverse groups are considered here, should career stage also be included?

ANSWER:

Unfortunately, we do not have career stage data available for this study.

- How do people become involved in GA4GH?

ANSWER:

The GA4GH website (<https://www.ga4gh.org/>) is the ideal place to learn about their mission, goals, and ongoing projects. Here one can learn the areas of genomics and health that GA4GH focuses on, such as data sharing, privacy, ethics, and interoperability.

The specific process of getting involved in GA4GH may vary depending on background, interests, and the current needs of the organization. Networking, active engagement, and a genuine commitment to GA4GH's mission can greatly increase chances of becoming involved in this global genomics and health community.

- Is anglosphere a commonly used term? Is the point that these are high income countries of largely white/European background?

ANSWER:

The term "Anglosphere" is commonly used to refer to a group of English-speaking countries that share historical, cultural, and political ties due to their common English language heritage. While the term is not universally known or used in all contexts, it is recognized and employed in various academic, geopolitical, and cultural discussions.

- The paper is reporting on outcomes of workshops and consultation etc, but I think there is a risk of speaking in such broad terms that it seems you don't consider the challenges in achieving EDI goals. I would find the paper considerably more useful if some increased specificity could be added, such as specific examples of what some of these proposed steps might be. I think this would then also allow for some consideration of the challenges of EDI goals to be discussed. I think this would take this from being a very general, "we need to be more equitable" paper to something that attempts to provide real guidance.

ANSWER:

We thank Reviewer 2 for this comment. In this revised version we have endeavoured to be a lot more specific in terms of the steps and challenges to be discussed in order to be of value to a wider audience. For instance, we have added a new table (Table 1) showing the community-pillars adopted by the EDI Advisory Group. We hope this adds more clarity to our lists. We have created Table 3 to emphasise clarity on steps towards adoption of EDI awareness and practice through 'Three Calls to Action'.

- Likewise, some of the suggested steps, like "Engage in outreach practices" (page 9) are easy to suggest but much more difficult to implement. As above, more specificity about how the groups think this might actually be achievable could be really powerful.

ANSWER:

This relates to the activities that have been described as part of the EDI Advisory Group projects. We agree that there were some unspecific activities there which are difficult to measure and difficult to assess. Because of that we have opted to delete those statements that we thought too unspecific. In particular, we have edited out the "Engage in outreach practices" and "Tactical EDI Issues" among others, which on reflection, clutter our message here. We hope that in this context 'more is less' and that what is left in Table 2 strengthens our argument there.

- Given the difficulties, there is very brief mention of how EDI goals might be evaluated, but considering the significant investment in time/resources being proposed (because truly achieving equity, diversity and inclusion really does require significant effort), surely these are a key part of such working group activities?

ANSWER:

We have included as a limitation the evaluation of EDI initiatives. We recognize this is essential to understanding their impact on the community, but recording measurements within our community remains a challenge and an active area of work which we hope to develop in the near future.

We would like to thank again our Reviewers for their valuable contributions and for the time and thinking that they have shown through their insightful comments. We believe that as a result of these we have a significantly improved version of our first submitted draft.

Referees' report, second round of review

N/A

Authors' response to the second round of review

N/A